# Concentration of 1,3-dimethyl-2-imidazolidinone in Aqueous Solutions by Sweeping Gas Membrane Distillation: From Bench to Industrial Scale

**DOI:** 10.3390/membranes9120158

**Published:** 2019-11-26

**Authors:** Ricardo Abejón, Hafedh Saidani, André Deratani, Christophe Richard, José Sánchez-Marcano

**Affiliations:** 1Institut Européen des Membranes UMR 5635, CNRS, ENSCM, Université de Montpellier, CC 047, Place Eugène Bataillon, 34095 Montpellier, France; ricardo.abejon@unican.es (R.A.); yahafedh@gmail.com (H.S.); andre.deratani@umontpellier.fr (A.D.); 2Department of Chemical and Biomolecular Engineering, University of Cantabria, Avda. Los Castros s/n, 39005 Santander, Cantabria, Spain; 3Kermel, 20 Rue Ampère, CEDEX, 68027 Colmar, France; christophe.richard68@wanadoo.fr

**Keywords:** sweeping gas membrane distillation, 1,3-dimethyl-2-imidazolidinone, solvent dehydration, hollow-fiber membrane, multi-objective optimization

## Abstract

Sweeping gas membrane distillation (SGMD) is a useful option for dehydration of aqueous solvent solutions. This study investigated the technical viability and competitiveness of the use of SGMD to concentrate aqueous solutions of 1,3-dimethyl-2-imidazolidinone (DMI), a dipolar aprotic solvent. The concentration from 30% to 50% of aqueous DMI solutions was attained in a bench installation with Liqui-Cel SuperPhobic^®^ hollow-fiber membranes. The selected membranes resulted in low vapor flux (below 0.15 kg/h·m^2^) but were also effective for minimization of DMI losses through the membranes, since these losses were maintained below 1% of the evaporated water flux. This fact implied that more than 99.2% of the DMI fed to the system was recovered in the produced concentrated solution. The influence of temperature and flowrate of the feed and sweep gas streams was analyzed to develop simple empirical models that represented the vapor permeation and DMI losses through the hollow-fiber membranes. The proposed models were successfully applied to the scaling-up of the process with a preliminary multi-objective optimization of the process based on the simultaneous minimization of the total membrane area, the heat requirement and the air consumption. Maximal feed temperature and air flowrate (and the corresponding high operation costs) were optimal conditions, but the excessive membrane area required implied an uncompetitive alternative for direct industrial application.

## 1. Introduction

Membrane distillation (MD) is a thermally driven process in which only vapor molecules are transported through porous hydrophobic membranes. The liquid feed to be treated by MD must be in direct contact with one side of the membrane, but without wetting the membrane to avoid the entrance inside the dry pores [1]. This hydrophobic nature of the membrane prevents the mass transfer in liquid phase and creates a vapor–liquid interface at the pore entrance (Figure 1). In this interface, the volatile compounds in the liquid feed evaporate and diffuse across the membrane pores. On the opposite side of the membrane, the vapor is condensed or removed, depending of the configuration of the MD system [2].

The benefits of MD compared to distillation or other separation processes based on membranes must be highlighted [3]: the complete theoretical rejection of ions, macromolecules, colloids, cells, and other non-volatiles; lower operating temperatures than conventional distillation; lower operating pressures than conventional pressure-driven membrane separation processes; reduced chemical interaction between membrane and process solutions; less demanding membrane mechanical property requirements; and reduced footprint spaces compared to conventional distillation processes.

The driving force at the origin of the mass transfer through the membrane is a partial pressure gradient, but this gradient can be induced and maintained by different mechanisms, which define the configuration of the MD system. In the case of sweeping gas membrane distillation (SGMD), a cold inert gas sweeps the permeate side of the membrane and removes the vapor molecules, which condensate outside the membrane module (Figure 2). SGMD presents some specific advantages when compared to other MD configurations, like relatively low heat loss by conduction through the membrane, low resistance to mass transfer in the gas phase, and a high driving force for transmembrane transport due to the continuous removal of vapor from the permeate side of the membrane [4].

The membranes are key elements for an effective implementation of SGMD systems. Hydrophobic mesoporous–macroporous membranes (pore diameters between 10 nm and 0.5 μm) made of polymeric materials such as polypropylene, polyethylene, polytetrafluoroethylene or polyvinylidenefluoride are generally selected for this application. As previously commented, the hydrophobic character is crucial to avoid the penetration of the aqueous phase through the membrane’s porosity. This characteristic can be estimated indirectly from the determination of the intrusion pressure. The general influence of other structural characteristics of the membrane on the performance of SGMD processes is shown in Table 1.

Three important parameters of the membrane have direct influence on the evaporative flux: the thickness of the membrane, the porosity, and the size of the pores. First of all, it is clear that the membrane resistance to mass transfer is proportional to the membrane thickness, so the thicker the membrane, the lower the flux. Moreover, it should be noted that the higher the porosity, the higher the flux. Indeed, a membrane having a high porosity offers a greater exchange surface area to the mass transfer and improved diffusivity [5]. Similarly, the evaporative flux increases with the pore diameter of the membrane. However, in order to avoid the penetration of liquid into the membrane, the pore diameter should not be too large. An optimum pore diameter value must therefore be determined for each SGMD application for a good compromise between performance and operation.

The influence of several operating variables on the SGMD processes has been previously investigated [6,7]. The temperatures of the feed and sweeping phases have been identified as the most relevant process conditions; in particular, the temperature difference between the two phases must be taken into consideration. Indeed, SGMD is a thermal process governed by the gradient of partial pressure induced by the difference of temperature on both sides of the membrane. Therefore, the greater the temperature difference between the two phases, the higher the evaporation flux. In addition, the flow velocities of the phases can have an effect on the performance, but it depends on the specific cases [5,8,9]. Table 2 summarizes the influence of the operation conditions on the evaporative flux in the SGMD processes.

Desalination has been the most investigated application of MD. This hybrid technology can be employed for the removal of salts and other undesired compounds from a saline water solution to produce freshwater, with quality enough for human consumption, agriculture or industrial uses [10]. Consequently, the development of an improved membrane for this particular application emerges as a very relevant hot topic of research [11,12]. Nevertheless, MD has been successfully implemented in other applications. For example, references of the employment of MD for the dehydration of different solutions, such as aloe vera juice [13], aqueous solutions of glycerol [14] or diethylene glycol [15], can be found in the bibliography. More specifically, SGMD has been applied for the recovery of volatile chemicals like aroma compounds, [16] but examples of its application to the dehydration of glycerol [17] and triethylene glycol [18] have been published as well.

Taking into account this available information, the main aim of this work is the analysis of the potential of SGMD for the concentration of aqueous solutions of 1,3-dimethyl-2-imidazolidinone (DMI). This chemical is a dipolar aprotic solvent with characteristics between tetrahydrofuran (THF) and hexamethylphosphoramide (HMPA), suitable for many types of organic reactions (especially organometallic reactions) [19,20,21,22,23,24,25,26,27]. Moreover, DMI is used in the manufacture of polymers, as well as in the industry of detergents, dyestuffs, and electronic materials. The recycling of this solvent from aqueous solutions without significant losses is very important for the sustainability of industrial processes. In this work, the concentration from 30% to 50% of aqueous DMI solutions was investigated, paying attention to the main variables of the process: temperatures and flowrates of the feed and sweep gas streams. From these experimental data, simple empirical models were developed to simulate the performance of the installation and calculate the vapor permeation and the DMI losses through the membranes. These empirical models can imply relevant simplification when compared with more complex models developed to represent the performance of SGMD. Finally, the preliminary scaling-up of the process was covered under different multi-objective optimization frameworks.

## 2. Experimental

### 2.1. Chemicals

DMI (99.9% of purity) was supplied by Kermel. Ultrapure water (>18.0 MΩ·cm resistivity) was obtained by a Milli-Q Element (Merck KGaA, Darmstadt, Germany).

### 2.2. Membrane Selection

The selection of a membrane for concentration by SGMD is strictly related to the intended application, as it depends on the nature of the solution to be concentrated. Among the SGMD membranes available on the market, the Liqui-Cel SuperPhobic^®^ membrane (provided by Alting, Hoerdt, France) appears to be the most suitable, since it is made of polypropylene, an inert polymer stable in contact with DMI. The membrane is hydrophobic and intended for applications with solutions characterized by surface tension values between 20 dyne/cm and 40 dyne/cm (in the case of DMI, its value is 40 dyne/cm). The membrane is commercially available as tubular modules of hollow fibers, in several different geometries. The module Liqui-Cel SuperPhobic^®^ X50 was selected and the characteristics of its fibers can be found in Table 3. The total membrane area of the module was 1.2 m^2^.

### 2.3. Experimental Installation

On the basis of the characteristics of the module described in the previous section, a SGMD bench installation was designed (Figure 3). Due to the nature of the solution to be concentrated, all of the equipment employed in the bench installation was made of stainless steel in order to be compatible with DMI.

As shown in Figure 3, the counter current configuration was selected. The DMI solution was stored in a 6 L stainless steel tank (5). During the operation, the volume inside the tank was controlled by a level sensor (12) (Immersion Piezo-Resistive Pressure, Keller 46X). The temperature of the solution was self-regulated by a heat exchanger consisting of a dipping coil (stainless steel), connected to a circulating cryo-thermostat (11). The solution circulation to the module was provided by a gear pump (1) (Micropump M520513), equipped with a speed controller (20–600 L/h). The flow rate was measured by a flow meter (2) with a micro-oval counter (Oval LSN45 LO). The solution passed through the module (6) outside the fibers and was recirculated back to the feed tank after leaving the module. The temperature and pressure were measured at the inlet and outlet of the module by a piezo-resistive pressure transmitter (0–5 bar) and a thermocouple (4), respectively. These same sensors were also installed at the inlet and outlet of the dry air compartment to measure the flow of air circulated in countercurrent inside the fibers. The relative humidity of the air (% RH) was measured at the inlet and outlet of the module by a humidity transmitter (7) (Delta Ohm HD2007) equipped with a remote sensor. Before entering the module, the air from the compressor (10) (7% RH) passed through a desiccant cartridge (9). The airflow was measured by a rotameter (8), in the range of 4–80 L/min (Brooks GT-1000). The airflow was regulated by means of the V6 control valve.

Depending on the position of the valves V1 to V4, the flows of the solution and the air can be done either in closed loop (without passing through the module) or in open loop (passing through the module). This makes it possible to stabilize the operating parameters (temperature, pressure, RH) before starting the concentration step. The V5 valve allows the tank to be drained. Each sensor was connected to a digital display to view the value in real time. The displays were assembled in a box that also provides power to the entire system.

### 2.4. Experimental Procedures

After the installation and testing of all the capabilities of the bench installation, the characterization of the system performance was carried out by determining its productivity in terms of evaporative flow through the membrane. These initial experiments were carried out by circulating ultrapure water in countercurrent with dry air (1.7% < RH < 5.0%). The resulting productivity was determined under different conditions by varying several parameters (water stream temperature and flowrate, air stream temperature and flowrate), to characterize the most relevant operation conditions. Each time a parameter was varied, the rest of the parameters were kept constant to analyze the influence of each variable individually. The evaporative flow was calculated from the difference in the measured relative humidity in the air stream before and after its pass through the module.

In the case of the concentration of DMI solutions, the experimental procedure was similar to that described for the tests with ultrapure water, with enough stabilization time to attain steady-state conditions. Based on the results of the analysis of the system with water, the operation conditions were chosen to obtain a high yield in terms of evaporative flow and, consequently, a high concentration rate of the DMI solution. The implementation of a valid method for the determination of the DMI concentration was essential for monitoring the performance of the concentration process of the aqueous solution. Moreover, any loss of DMI across the membrane during the concentration process must be identified and quantified. The concentration of DMI solutions was measured by refractometry (the evolution of the refractometer signal as a function of the DMI concentration for a range of 40–3000 ppm was lineal). Therefore, the proposed assay method was validated with a detection limit of 40 ppm.

## 3. Results and Discussion

### 3.1. Pure Water Permeation

#### 3.1.1. Determination of the Stabilization Time

The determination of the time required to attain steady-state conditions was the first task to be completed. These experiments were carried out by fixing all parameters of the process (temperature, pressure, water and air flowrates shown in Table 4) and measuring the evolution of the water evaporation flux until stabilization.

The water vapor flux measured increases (+5.6%) during the first two hours of operation, reaching 0.09 kg/h·m^2^. Then, the evaporation flow after this time until 5 h of operation did not show additional increases, as the values of the evaporative flow obtained were ranged within a margin of error of 3%, which could be attributable to the measurement error of the humidity sensors. According to these results, the stabilization time was estimated to be 2 h. In the following experiments, each measurement point corresponds to the flow value determined after at least a stabilization time of 2 h.

#### 3.1.2. Analysis of the Influence of the Water and Air Temperatures

SGMD is a thermal process governed by a partial pressure gradient as driving force. This gradient is induced by the temperature difference between both compartments separated by the membrane. Therefore, the greater the temperature difference between the two phases, the greater the evaporative flow. This fact justifies the analysis of the influence of the temperature of the water and air streams entering the membrane module.

Since the maximum operating temperature tolerated by the membrane module was 70 °C, the variation of the evaporative flow as a function of the water temperature was investigated in the range of 20 °C to 67 °C (Figure 4). The operation conditions of these tests are equivalent to the ones summarized in Table 4, but with variation of the temperature of the streams.

As shown in Figure 4, the evaporative flow increased with increasing water temperature according to an exponential law under constant air temperature (21 °C). For example, a flux rise of 365% was recorded between 40 °C and 67 °C. This evolution is directly linked to the increase of water vapor pressure as a function of the water temperature which corresponds to the water vapor enhancement described by the Antoine equation [28]. Although the bench installation was not designed to control the temperature of the air stream, the system was prepared to allow the measurement of the evaporative flow at two different inlet air temperatures (21 °C and 26 °C). Under constant water temperature conditions (60 °C), the decrease of air temperature increased the temperature difference between the water and the air, thus increasing the evaporative flow. The variation in air temperature should be considered to be as important as that of the water temperature, since a decrease in the air temperature from 26 °C to 21 °C increased the evaporative flow by 36%.

#### 3.1.3. Analysis of the Influence of the Water and Air Flowrates

According to Khayet [29], the water flowrate at the inlet of the module has practically no effect on the evaporative flow. During the preliminary tests to characterize the bench unit, some experiments were carried out to confirm the insignificant incidence of the water flowrate on the system performance. The obtained results (not shown) confirmed this hypothesis and the same conclusion resulted from experiments with DMI solutions, where the influence of the solution flowrate on the separation process was negligible.

The analysis of the influence of the air flowrate on the evaporative flow revealed it strongly depended on other variables: the temperature of the water appeared as the most relevant operation condition. Therefore, the evaporative flow was determined as a function of the air flowrate at different water temperatures. These operation conditions are summarized in Table 5.

The evolution of the evaporative flow as a function of the air flow at different water temperatures is shown in Figure 5. As shown, the evaporative flow was only slightly dependent on the air flowrate, except for the cases where the highest water temperatures were reached. Indeed, in the water temperature range between 20 °C and 50 °C, the evaporative flux was practically constant even if the air flowrate was increased from 20 L/min to 34 L/min.

Under the low water temperature conditions (temperatures below 40 °C), an evaporative flow saturation value was reached at 20 L/min. When the feed temperature was increased to 67 °C, the saturation stage is shifted at higher air flow rates. Thus, an increase in evaporative flow is recorded with the air flow rate. The increase is 32% when the air flow rate increases from 25 L/min to 34 L/min (limit of the air flowrate available through the installation).

### 3.2. Concentration of DMI Solutions

#### 3.2.1. Analysis of the Influence of the Initial DMI Concentration

The effect of the composition of the feed aqueous solution was studied by reporting the evolution of the evaporative flow as a function of the DMI concentration in the feed solution. The operating conditions of these tests are compiled in Table 6.

The DMI concentration was varied in the range of 0–33%, and it was shown that the increasing addition of DMI to the feed solution had no effect on the evaporative flow, as the flow maintained a constant value of 0.11 kg/h·m^2^.

#### 3.2.2. Analysis of the Losses of DMI by Permeation through the Membrane

During dehydration of an aqueous DMI solution by SGMD, only the water vapor is assumed to pass through the hydrophobic membrane. However, DMI can permeate through the membrane just by steam drag.

In order to detect and quantify any loss of DMI, the air at the outlet of the module was condensed in an immersion trap with liquid nitrogen. The condensate was recovered every 10 min, which made it possible to have a sufficient quantity for DMI quantification. Therefore, at the end of the experiment, the condensed evaporative flux is determined by weighing. This condensed evaporative flux corresponded to a fraction of the total evaporative flux, which was determined by calculation from the registered relative humidity values. Two different sets of experiments were carried out with changes in the DMI compositions, one with 10% and the other with 33%. The total losses of DMI depended on the concentration of the feed solution, since the higher losses corresponded to the more concentrated solution: while loss values of 0.120% were assessed for a 10% DMI feed solution in the reservoir, the losses increased to 0.195% when a 33% DMI feed solution was employed.

#### 3.2.3. Achievement of Concentrated DMI Solutions

The final aim of the experimental tests consisted of the implementation of a concentration test according to a defined specification: attain an aqueous solution with 50% DMI concentration starting from a solution of approximately 30% DMI. The operation conditions for this concentration test are summarized in Table 7.

The evolution of the evaporative flux as a function of the time of the experiment was recorded (not shown). A slight variation of the flux (4.5% reduction from the initial to the final sample) was observed. From this hourly-determined register of the evaporative flux, the evolution of the DMI concentration in the reservoir throughout the experiment was calculated and compared with the samples taken (Figure 6). The target concentration (50%) was reached after 9 h of operation.

The assessment of the amount of DMI that escaped from the module was possible. The DMI loss was expressed as the cumulative mass percentage of DMI in the total condensate mass. Figure 7 shows the evolution of DMI loss as a function of the DMI concentration in the tank, which increased according to an exponential law. It was observed that, at the end of the experimental test (DMI concentration equal to 50%), the cumulative loss was close to 0.55% (this is less than 6 mL per L of evaporated water). This result indicates the possibility to concentrate DMI solutions in spite of the very low vapor permeation flux, which is certainly low for an industrial process if we compare with classical values of membrane distillation processes for water desalination or concentration of salty solutions in literature (from 0.1–10 kg/h·m^2^). However, we have to take in consideration that there are not losses of solute in such processes of salty solutions distillation. On the contrary, in the case of the DMI concentration, a critical point concerns DMI losses. These losses are minimized by using a Liqui-Cel SuperPhobic^®^ X50 module which contains hollow fibers with only 40% porosity and 0.04 µm mean pore size. These characteristics are responsible for both the low vapor flux observed and low DMI losses.

### 3.3. Process Modeling and Scaled-Up Design

Once the technical viability of the process for the concentration of aqueous solutions of DMI by SGMD was demonstrated, the preliminary design of a scaled-up installation was carried out. Although highly robust two-dimensional theoretical models for SGMD simulation have been developed [4,30,31,32], several authors have preferred the application of empirical models because of their higher simplicity [8,9].

In this work, an empirical model was proposed to describe the performance of the process as a function of the main operation variables: feed inlet temperature *T*, air flowrate *V_AIR_*, and DMI concentration *C_DMI_*. The performance of the SGMD system was characterized by the assessment of the fluxes of the two components (water and DMI). On the one hand, the water flux *F* was calculated with the following equations:
(1)$$F=F_{MAX}\cdot \frac{e^{\frac{V_{AIR}}{V_{E}}}-e^{-\frac{V_{AIR}}{V_{E}}}}{e^{\frac{V_{AIR}}{V_{E}}}+e^{-\frac{V_{AIR}}{V_{E}}}}$$
(2)$$F_{MAX}=F_{0}\cdot e^{\alpha \cdot T}$$
(3)$$V_{E}=V_{0}\cdot e^{\beta \cdot T}$$
where *F_MAX_* is the maximal *F* value that can be attained at one temperature, *V_E_* is the parameter that determines the *V_AIR_* influence on the *F* value (when *V_AIR_* equals *V_E_* the *F* value is 0.762·*F_MAX_*), *F*_0_ and *V*_0_ and *α* and *β* are the corresponding baseline values and exponential parameters of the temperature dependence of the *F_MAX_* and *V_E_* values, respectively. On the other hand, the DMI flux *F_DMI_* was calculated with the following equations:
(4)$$F_{DMI}=F\cdot \frac{L_{DMI}}{100}$$
(5)$$L_{DMI}=L_{0}\cdot e^{\omega \cdot C_{DMI}}$$
where *L_DMI_* represents the percentage of DMI losses through the membrane and *L*_0_ and *ω* represent the baseline value and exponential parameter of the concentration dependence. The values of all the parameters required by the proposed model were obtained after minimization of the errors in the fitting of the experimental data and they are compiled in Table 8. The satisfactory fitting of the experimental data by the proposed model can be observed in Figure 4, Figure 5 and Figure 7.

Nevertheless, before the preliminary design of a scaled-up installation was done, a complete sensibility analysis of the water flux through the membrane as a function of the feed temperature and the air flowrate was carried out (Figure 8). As the figure shows, the influence of the feed temperature on the performance of the process was more important that the influence of the air flowrate. On the one hand, once certain air flowrate points were attained, values besides these limits resulted in a plateau area where it was not possible to significantly increase the evaporation flux. On the other hand, the exponential influence of the feed temperature resulted in highly improved evaporation fluxes were the maximal considered temperature values were applied. The maximal evaporation flux value was 0.141 kg/h·m^2^, which corresponded to 70 °C and 80 L/min, but a value equal to 0.136 kg/h·m^2^ was obtained with 70 °C and less than 60 L/min.

The preliminary design of a scaled-up installation (based on the use of the membrane modules experimentally characterized in this work) was able to concentrate 3400 kg/h of a 30% DMI solution to achieve a final concentration of 50% DMI. According to the proposed model and the corresponding calculated DMI losses, the installation must be able to evaporate 1368 kg/h of water, which implied 8 kg/h of DMI lost through the membrane. Indeed, 2024 kg/h of a 50% DMI concentrated solution can be obtained.

The optimal design of the installation is not a simple task because of the existence of contradictory objectives. Although the main objective of the design should be the minimization of the total membrane area required, this situation implied maximal feed temperature and air flowrate and, consequently, maximal operation costs. Therefore, 9635 m^2^ was the minimal membrane area of the installation when the maximal air flowrate (80 L/min) and feed temperature (70 °C) were applied. The complete evolution of the membrane area required in the installation as a function of the selected air flowrates and feed temperatures can be observed in Figure 9.

In order to have a simplified outlook to the multi-objective optimization of the installation, Pareto graphs were prepared. Pareto optimal solutions are solutions that cannot be improved in one objective function without deteriorating the performance in at least another objective [33]. In this case, the epsilon constraint method was employed to obtain Pareto graphs that included the total membrane area compared to total air consumption or total heat requirements [34,35].

Total air consumption was selected as the most adequate variable to take into account the costs due to the selected air flowrate value. The Pareto that resulted from the consideration of the simultaneous minimization of the total membrane and the air consumption is graphed in Figure 10. The results clearly demonstrated that air consumption values above 350 m^3^/min only slightly decreased the total membrane area, so air flowrate values below the maximal value analyzed in this work could be selected without excessive detriment in the process performance.

However, this is not the case for the feed temperature and the corresponding total heat requirement, which was calculated by considering the heat capacity of pure DMI (1.8 J/g·°C) and 60% efficacy in the heat exchanger [36,37]. As shown in Figure 11, the reduction of the heat requirement (by selection of lower feed temperature) implied great membrane area penalties. Therefore, the installation should be designed to work at the maximal feed temperature. Nevertheless, even under these optimal conditions, the resulted total membrane area is excessive. In these circumstances, the scaled-up process cannot be considered competitive and further additional work will be required. These future tasks should consider the identification of more adequate membrane modules, maybe with increased vapor flux, although this should imply increased DMI losses. The design of more complex configurations, with in series stages to recover DMI from the sweep phase, should be taken into account.

## 4. Conclusions

This work investigated the applicability of sweeping gas membrane distillation (SGMD) to concentrate by dehydration aqueous solutions of 1,3-dimethyl-2-imidazolidinone (DMI). An experimental bench installation equipped with Liqui-Cel SuperPhobic^®^ membranes was employed to analyze the influence of the main operation variables on the process performance. This way, the temperature of the feed stream and the air flowrate were identified as the most relevant variables.

The installation was successfully employed to achieve a concentration of DMI solutions from 30% to 50% under batch conditions. The selected membranes were responsible for the low vapor flux observed but were also effective for the minimization of DMI losses through the membrane since these losses were maintained below 1% of the evaporated water flux. This fact implied that more than 99.2% of the DMI fed to the system was recovered in the produced concentrated solution.

Once the technical viability of the process was confirmed, simple empirical models were developed to simulate the performance of the SGMD process for DMI concentration. These models were applied to the design of a scaled-up installation able to concentrate 3400 kg/h of a 30% DMI solution under continuous operation. The analysis of the influence of the main operation variables was taken into consideration to have a preliminary multi-objective optimization of the system by simultaneous minimization of the total membrane area, the heat requirement and the air consumption. The simulation results show that maximal feed temperature and air flowrate (and the corresponding high operation costs), as well as very high membrane area (9635 m^2^) were necessary to reach the contradictory objectives of the process. Under these conditions, the process was deemed to not be competitive for an industrial application and it should be improved before a definitive cost analysis that could be very carefully compared with classical distillation processes.

## Figures and Tables

**Figure 1 membranes-09-00158-f001:**
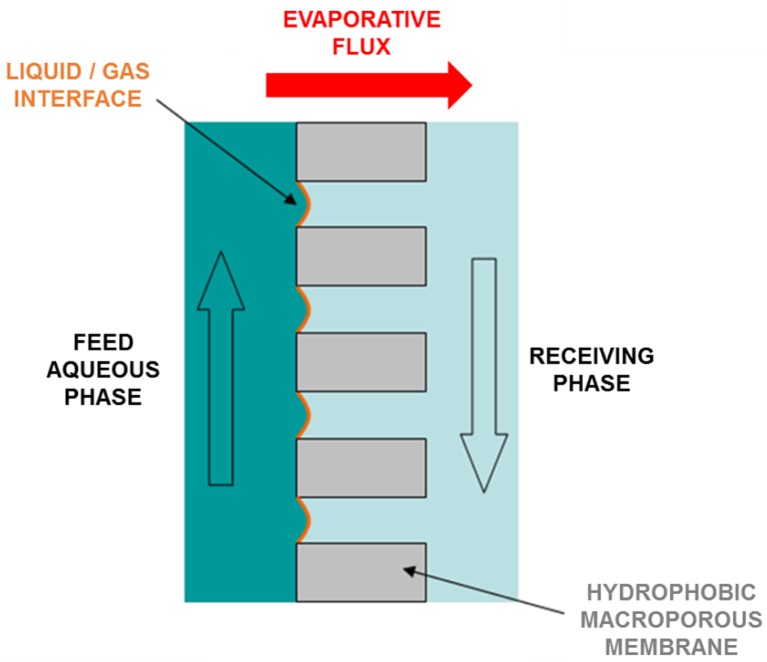
Schematic representation of the performance of membrane distillation.

**Figure 2 membranes-09-00158-f002:**
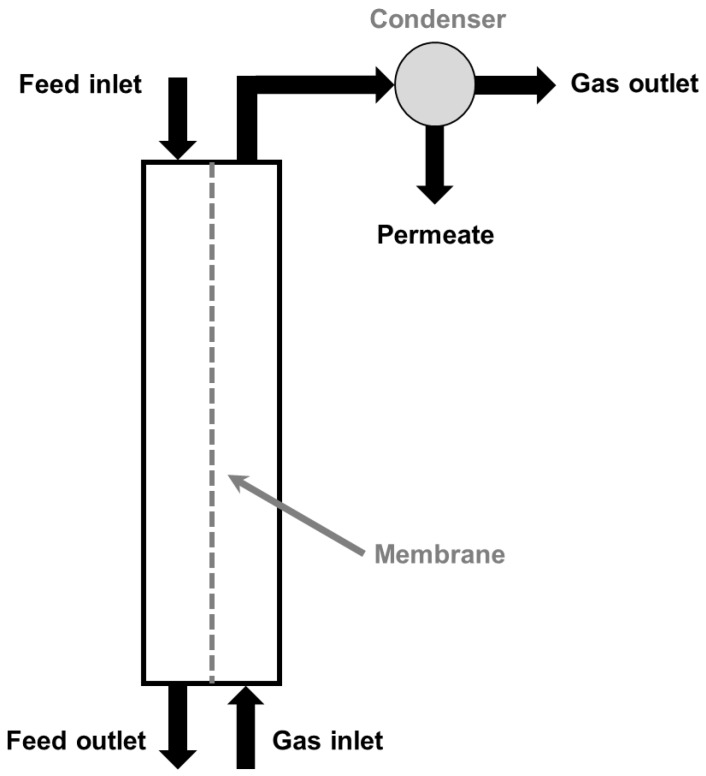
Configuration of a sweeping gas membrane distillation (SGMD) system.

**Figure 3 membranes-09-00158-f003:**
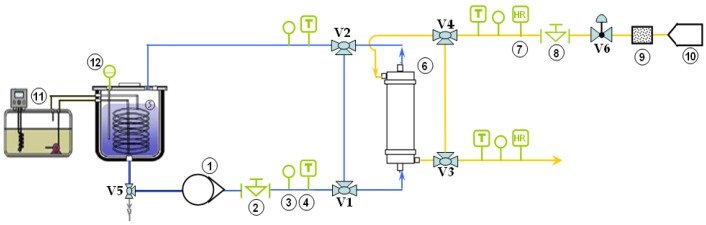
Scheme of the experimental SGMD bench installation.

**Figure 4 membranes-09-00158-f004:**
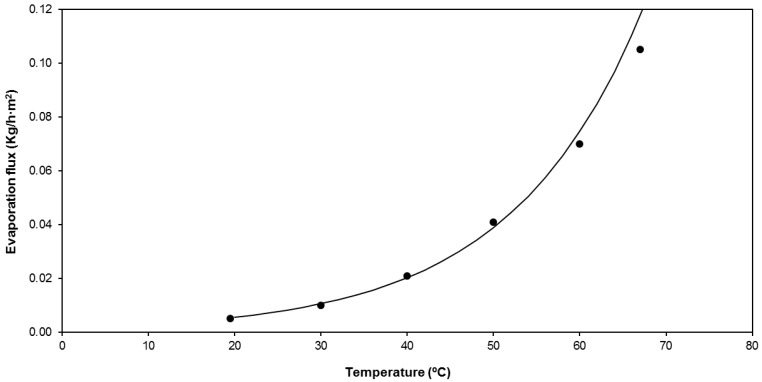
Variation of the evaporative flow as a function of the water temperature.

**Figure 5 membranes-09-00158-f005:**
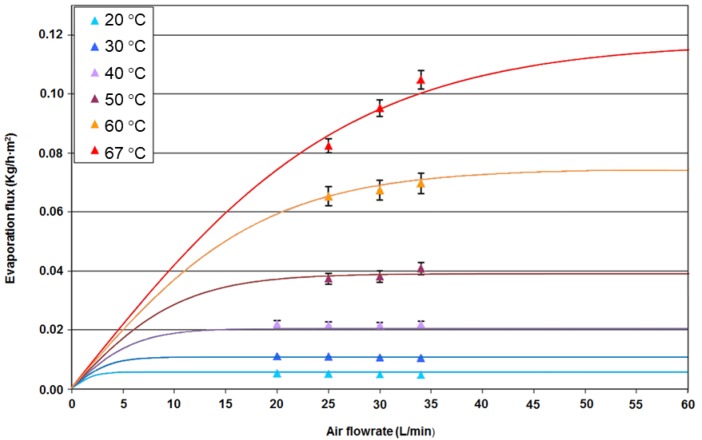
Variation of the evaporative flow as a function of the air flowrate for different water temperatures.

**Figure 6 membranes-09-00158-f006:**
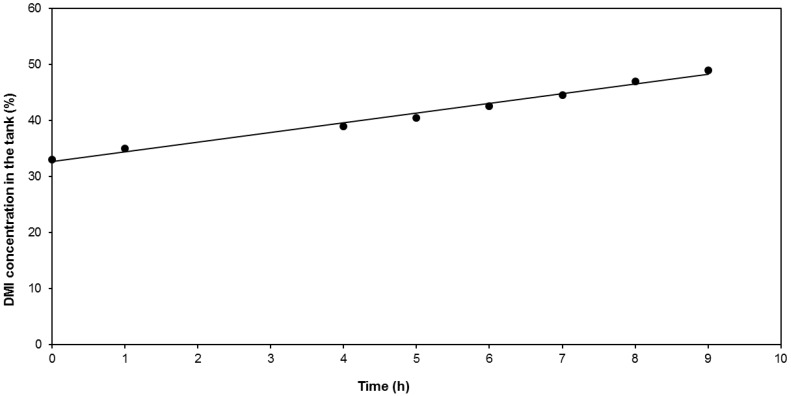
Evolution of the DMI concentration in the feed tank through the time.

**Figure 7 membranes-09-00158-f007:**
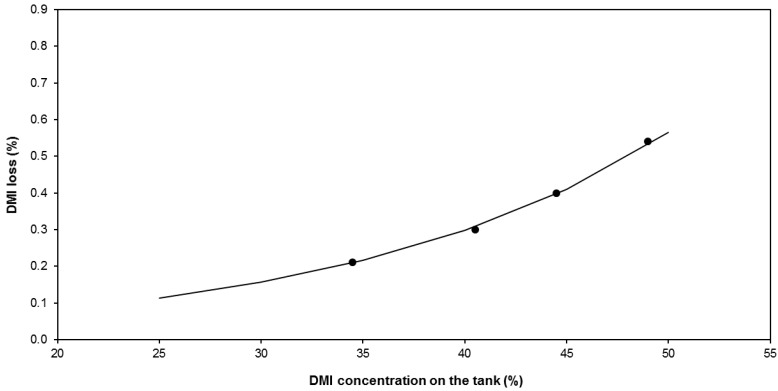
Variation of the DMI losses as a function of the DMI concentration of the feed solution.

**Figure 8 membranes-09-00158-f008:**
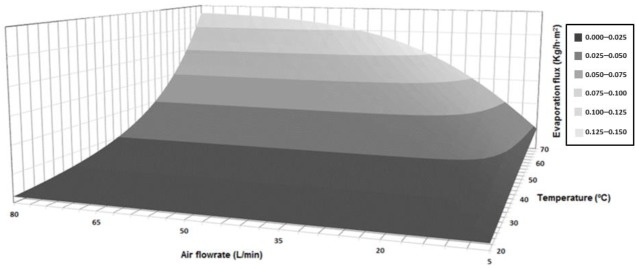
Simulated evaporation flux of the system under different feed temperature and air flowrate conditions.

**Figure 9 membranes-09-00158-f009:**
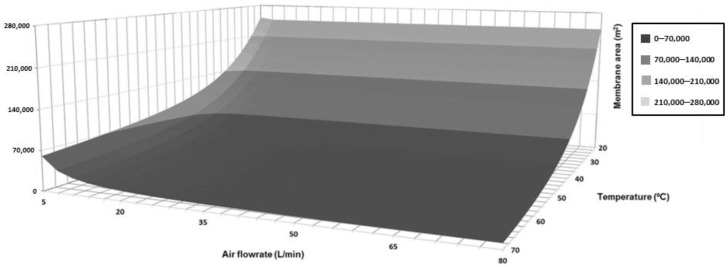
Evolution of the total membrane area required in the installation under different feed temperature and air flowrate conditions.

**Figure 10 membranes-09-00158-f010:**
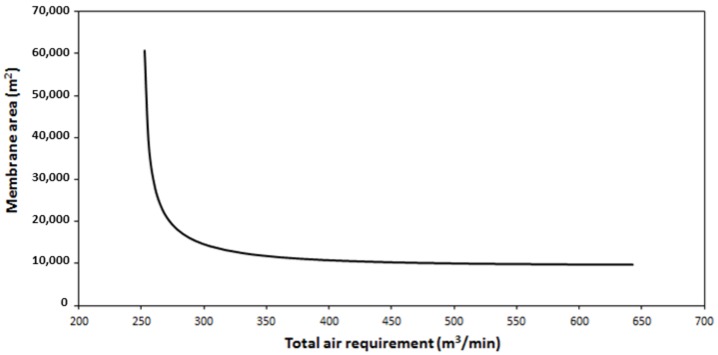
Influence of the total air requirement as sweeping gas on the total membrane area of the process.

**Figure 11 membranes-09-00158-f011:**
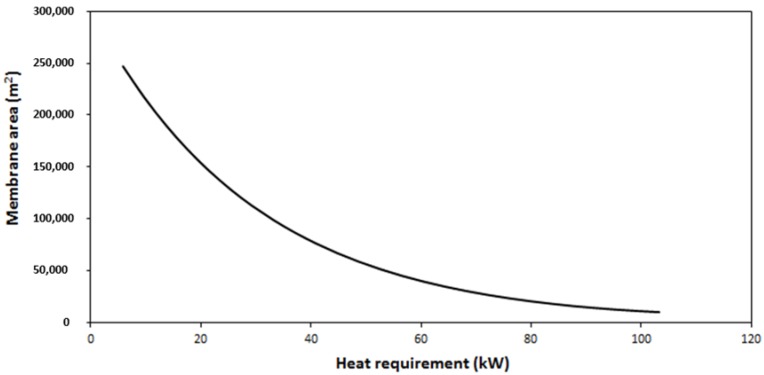
Influence of the heat requirement to increase the feed temperature on the total membrane area of the process.

**Table 1 membranes-09-00158-t001:** Influence of the membrane characteristics on the evaporative flux. + = positive effect; − = negative effect; / = non-referenced effect.

Membrane Characteristics					
Thickness	Porosity	Pore Size	Pore Size Distribution	Tortuosity	Surface Geometry
− − − −	+ + + +	+ + + +	/	− −	/

**Table 2 membranes-09-00158-t002:** Influence of the operation conditions on the evaporative flux.

Operation Conditions					
Feed Side			Sweeping Side		
Temperature	Flowrate	Flow Regime	Temperature	Flowrate	Flow Regime
+ + + +	+ +	/	− −	+	/

**Table 3 membranes-09-00158-t003:** Characteristics of the hollow fibers in a Liqui-Cel SuperPhobic^®^ X50 module.

Material	Outer Diameter (µm)	Inner Diameter (µm)	Bubble Point (psi)	Porosity (%)	Pore Diameter (µm)
Polypropylene	300	220	240	40	0.04

**Table 4 membranes-09-00158-t004:** Conditions of the experiments for the assessment of the stabilization time. RH = relative humidity.

Water Inlet			Air Inlet				Time (h)
Temperature (°C)	Pressure (bar)	Flowrate (L/min)	Temperature (°C)	Pressure (bar)	Flowrate (L/min)	RH (%)	
60	1.05	3	21	0.9	30	1.7	1–5

**Table 5 membranes-09-00158-t005:** Conditions of the experiments for the analysis of the influence of the water and air flowrates.

Water Inlet			Air Inlet			
Temperature (°C)	Pressure (bar)	Flowrate (L/min)	Temperature (°C)	Pressure (bar)	Flowrate (L/min)	RH (%)
20–67	1.05	3	25	0.9	20–34	1.7

**Table 6 membranes-09-00158-t006:** Conditions of the experiments for the analysis of the influence of the initial 1,3-dimethyl-2-imidazolidinone (DMI) concentration (*C_DMI_*).

Water Inlet			Air Inlet				*C_DMI_* (%)
Temperature (°C)	Pressure (bar)	Flowrate (L/min)	Temperature (°C)	Pressure (bar)	Flowrate (L/min)	RH (%)	
67	1.05	3	25	0.9	34	1.7	0–33

**Table 7 membranes-09-00158-t007:** Conditions of the concentration test to obtain a concentrated (50%) DMI solution.

Water Inlet			Air Inlet				Time (h)
Temperature (°C)	Pressure (bar)	Flowrate (L/min)	Temperature (°C)	Pressure (bar)	Flowrate (L/min)	RH (%)	
60	1.05	3	21	0.9	30	1.7	0–9

**Table 8 membranes-09-00158-t008:** Parameters of the developed SGMD model.

Parameter	Unit	Value
*F* _0_	kg/h·m^2^	0.0015
*V* _0_	L/min	0.7265
*L* _0_	%	0.0228
*α*	°C^−1^	0.0651
*β*	°C^−1^	0.0539
*ω*		0.0642

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
