# Peer review of "Concentration of 1,3-dimethyl-2-imidazolidinone in Aqueous Solutions by Sweeping Gas Membrane Distillation: From Bench to Industrial Scale"

_membranes, 2019, doi:10.3390/membranes9120158_

Round 1

Reviewer 1 Report

This manuscript investigated the parameters that influencing the concentration of DMI in SGMD in bench scale, and then designed the scale-up process by modeling. Comments are as follows:

The title of this article seems not quite precise to cover the whole content of the test, since it doesn’t show the scale-up modeling process. Please specify each name of different devices and machines in Figure 3. Please modify the grammar of this sentence in section 3.1.2 “Therefore, the greater is the temperature difference between the two phases, the greater is the evaporative flow.” Please correct all the “oC” to “oC”.

Author Response

Reviewer 1

This manuscript investigated the parameters that influencing the concentration of DMI in SGMD in bench scale, and then designed the scale-up process by modeling. Comments are as follows:

The title of this article seems not quite precise to cover the whole content of the test, since it doesn’t show the scale-up modeling process.

The title has been modified.

Please specify each name of different devices and machines in Figure 3.

The devices and machines in Figure 3 are now described in the text.

Please modify the grammar of this sentence in section 3.1.2 “Therefore, the greater is the temperature difference between the two phases, the greater is the evaporative flow.”

The sentence has been corrected.

Please correct all the “oC” to “°C”.

The symbols have been corrected.

Reviewer 2 Report

In the manuscript titled “Concentration of 1,3-dimethyl-2-imidazolidinone in aqueous solutions by sweeping gas membrane distillation”, the authors investigated the operation parameters of using SGMD to dehydrate aqueous solutions of DMI on a module. Although minimal losses of DMI was achieved. A scale-up analysis was carried out to found that the membrane being used in this study was not viable in scale-up because of the required membrane area. The authors presented sufficient background and discussions of the results. I suggest the manuscript to be published in Membranes with minor

Please correct the typos in Figure 3 and Table 4: HR as RH (Relative humidity); Pression as Pressure. In the experiment for determining stabilization time, the conditions chosen are among the extreme values. This raise the concern for other experiments carried at smaller temperature gradient or lower water/air flowrate may not be reaching steady state at 2 h. The stabilization time needs to be confirmed on a case-by-case basis, or at least be confirmed on a combination of small temperature gradient and low flowrate. The reason for the different dependence of evaporation flux on air flowrate at different water temperature is because the rate limiting step of the system changed at different temperature. At higher temperature, the downstream air flow of 25 L/min was not enough to remove the evaporated water vapor and make the effective partial pressure difference across the membrane become smaller. This can be improved with a higher air flow rate as the downstream water concentration is lower.

Author Response

Reviewer 2

In the manuscript titled “Concentration of 1,3-dimethyl-2-imidazolidinone in aqueous solutions by sweeping gas membrane distillation”, the authors investigated the operation parameters of using SGMD to dehydrate aqueous solutions of DMI on a module. Although minimal losses of DMI was achieved. A scale-up analysis was carried out to found that the membrane being used in this study was not viable in scale-up because of the required membrane area. The authors presented sufficient background and discussions of the results. I suggest the manuscript to be published in Membranes with minor

Please correct the typos in Figure 3 and Table 4: HR as RH (Relative humidity); Pression as Pressure.

The typos have been corrected.

In the experiment for determining stabilization time, the conditions chosen are among the extreme values. This raise the concern for other experiments carried at smaller temperature gradient or lower water/air flowrate may not be reaching steady state at 2 h. The stabilization time needs to be confirmed on a case-by-case basis, or at least be confirmed on a combination of small temperature gradient and low flowrate. The reason for the different dependence of evaporation flux on air flowrate at different water temperature is because the rate limiting step of the system changed at different temperature. At higher temperature, the downstream air flow of 25 L/min was not enough to remove the evaporated water vapor and make the effective partial pressure difference across the membrane become smaller. This can be improved with a higher air flow rate as the downstream water concentration is lower.

A comment about the stabilization time has been included.